# Effects of a Serratus Anterior Plane Block After Video-Assisted Lung Wedge Resection: A Single-Center, Prospective, and Randomized Controlled Trial

**DOI:** 10.3390/medicina61010011

**Published:** 2024-12-26

**Authors:** Seokjin Lee, Tae-Yun Sung, Choon-Kyu Cho, Gyuwon Lee, Woojin Kwon

**Affiliations:** 1Department of Anesthesiology and Pain Medicine, Konyang University Hospital, Konyang University College of Medicine, Daejeon 35365, Republic of Korea; sj1825@kyuh.ac.kr (S.L.);; 2Myunggok Medical Research Institute, Konyang University Hospital, Konyang University College of Medicine, Daejeon 35365, Republic of Korea

**Keywords:** serratus anterior plane block, ultrasound, video-assisted thoracoscopic surgery, acute pain service, opioid-sparing anesthesia

## Abstract

*Background and Objectives*: Video-assisted thoracoscopic surgery (VATS) is associated with less postoperative pain than traditional open thoracotomy. However, trocar and chest tube placement may damage the intercostal nerves, causing significant discomfort. An ultrasound-guided serratus anterior plane block (SAPB) is a promising mode of pain management; this reduces the need for opioids and the associated side-effects. This study evaluated whether SAPB, compared to intravenous analgesia alone, reduces opioid consumption after thoracoscopic lung wedge resection. *Materials and Methods*: In total, 22 patients undergoing VATS lung wedge resections were randomized into two groups (SAPB and control): both received intravenous patient-controlled analgesia (PCA), and one group received additional SAPB. The primary outcome was the cumulative intravenous fentanyl consumption at 8 h postoperatively. The visual analog scale (VAS) pain scores and the incidence of postoperative complications were assessed over 48 h post surgery. *Results*: Fentanyl consumption by 8 h post surgery was significantly lower in the SAPB group than in the control group (183 ± 107 μg vs. 347 ± 202 μg, *p* = 0.035). Although the VAS scores decreased with time in both groups, the differences were not statistically significant. The SAPB group required fewer opioids by 48 h. No significant between-group differences were observed in postoperative complications, including nausea and vomiting. *Conclusions*: SAPB effectively reduced opioid consumption after VATS lung wedge resection. SABP may serve as a valuable component of multimodal pain management.

## 1. Introduction

Low-dose computed tomography has increased the early detection rate of lung cancer [1,2]. In contrast to advanced cancer, early stage cancer can often be adequately treated using less-invasive procedures such as segmentectomy or wedge resection employing video-assisted thoracoscopic surgery (VATS) [3]. The reduction in surgical incision size compared to that of conventional lobectomy decreases pain [4]. However, during thoracoscopic surgery, the trocar can damage the intercostal nerves. In addition, a postoperatively placed chest tube may cause pain during breathing [5]. The risk of progression to chronic postsurgical pain (CPSP) is high [6], and proactive postoperative pain management is crucial. Inadequate pain control can render it impossible to breathe deeply, triggering atelectasis. A pain-induced increase in sympathetic nervous system activity may cause hypertension and increase the risk of postoperative infection via immunosuppression.

After VATS lung wedge resection thoracic surgery, several methods can be used for postoperative pain management. The simplest is intravenous administration of opioid analgesics. However, pain relief using opioids often requires large doses; the side-effects include respiratory depression, nausea, vomiting, and itching [7,8]. Regional analgesic methods include thoracic epidural analgesia (TEA), a thoracic paravertebral block (TPVB), an intercostal block, and interpleural local anesthetic infusion [9]. TEA and TPVB are the most effective methods of pain control [10,11]. However, both methods are invasive and may be associated with complications such as pneumothorax, spinal cord injury, hypotension caused by a central nerve block, and/or respiratory issues attributable to muscle depression. Drugs delivered via intercostal catheters, and interpleural drug infusions, do not afford adequate pain control during VATS [12].

Less-invasive analgesia that nonetheless ensures effective pain relief in VATS lung wedge resection would enhance the surgical outcomes and reduce complications. The use of an ultrasound-assisted Serratus anterior plane block (SAPB) as an alternative to a neuroaxial block or a TPVB may afford effective lateral thoracic paresthesia and reduce the incidence of side effects [13]. A local anesthetic is delivered under ultrasound guidance to the lower or upper portion of the serratus anterior muscle. This blocks the lateral branch of the intercostal nerve, thereby ensuring analgesia of the lateral aspect of the rib cage [14]. The procedure is relatively simple and is associated with few complications. The anesthetic is injected under the serratus anterior muscle, which is readily identified via ultrasound; complications associated with TEB and TPVB are absent. Several studies have reported that SAPB ensures effective analgesia during breast surgery, thoracotomy, and minimally invasive valve surgery [15,16,17]. It also controls pain in patients with rib fractures and those who undergo other thoracic surgeries. For thoracotomy patients, SABP affords pain relief equivalent to that of TEA, and is associated with less hypotension [11].

Previous studies have focused on lung lobectomy. This major surgical procedure is associated with significant postoperative discomfort. Recently, research has expanded to include thoracoscopic surgery, reflecting the growing use of minimally invasive techniques. However, the increasing inclusion of a broader patient population, such as those undergoing lobectomy or segmentectomy via VATS, introduces additional variables.

In this study, we hypothesized that SAPB would reduce opioid consumption, as measured by patient-controlled analgesia (PCA) data, following thoracoscopic lung wedge resection, compared to the non-block group.

## 2. Materials and Methods

### 2.1. Patient Selection

This study was approved by the Institutional Review Board of Daejeon St. Mary’s Hospital, Daejeon, South Korea (approval no. DC17EESE0076, 30 October 2017), and was registered with the Clinical Research Information Service (code KCT0002626, https://cris.nih.go.kr/cris/search/detailSearch.do?search_lang=E&focus=reset_12&search_page=L&pageSize=10&page=undefined&seq=11875&status=5&seq_group=9639, accessed on 4 January 2018). The single-center prospective randomized controlled study ran from November 2017 to March 2019. Daejeon St. Mary’s Hospital is an academic hospital affiliated with the College of Medicine, The Catholic University of Korea. The hospital performs over 10,000 general anesthesia surgeries annually, and the Department of Thoracic Surgery conducts more than 80 VATS procedures each year. There are 12 operating rooms, supported by 7 anesthesiology faculty members and 18 nurse anesthetists. The hospital actively utilizes regional anesthesia techniques and takes a leading role in managing postoperative pain control. The personal information of the participants is accessible only to authorized collaborators with permission to access such data. The collected data are securely stored in a locked cabinet. Informed consent was obtained from patients and/or their guardians after a thorough explanation (at least 10 min) on the day before surgery.

### 2.2. Inclusion and Exclusion Criteria

The inclusion criteria were patients who underwent lung wedge resection and who requested PCA in our thoracic surgery department, age 20–70 years and American Society of Anesthesiologists (ASA) physical status 1–3. The exclusion criteria were ASA physical status 4 or higher, a history of chronic pain treatment or drug dependence, an inability to communicate with the researcher, current anticoagulant therapy or a bleeding disorder to prevent hematomas caused by vascular puncture, a body mass index (BMI) of 35 kg/m^2^ or higher, a lack of a PCA request, and refusal of nerve block. In addition, those who had bled more than 1 L during surgery, whose surgical times exceeded 6 h, and who required mechanical ventilation after surgery were excluded.

### 2.3. Randomization and Blinding

The patients were divided into an SAPB group and a control group with equal allocation. A computer-generated random number table was created using R software (version R 3.4.4), and the numbers were used to allocate all patients into one of the two groups. The numbers were placed in individual envelopes that were opened on the morning of surgery.

To reduce bias in the study, two individuals performed independent roles as the proceduralist and the evaluator. The proceduralist opened a sealed envelope containing the randomization assignment on the morning of the surgery to determine whether the participant would be in the control group or the SAPB group.

If the patient was in the control group, the proceduralist prepared the PCA medication by reference to the patient’s weight. If the patient was in the SAPB group, the proceduralist prepared both the PCA and the local anesthetic. To prevent incorrect mixing of the local anesthetic, a label “IRB number, SAPB group, for research use” was placed on the syringe. After surgery, the labeled syringe was used to establish the SAPB. The evaluator assessed all patients and recorded all data, including pain scores and opioid consumption.

### 2.4. Surgical Course

Anesthesia initiation and maintenance were identical for both groups; no premedication was administered. When a patient arrived in the operating room, noninvasive blood pressure, electrocardiograph, and oxygenation monitors were placed. Propofol 2 mg/kg and rocuronium 0.6 mg/kg were administered after preoxygenation. Once the loss of consciousness was adequate, and muscle relaxation was apparent, as confirmed by a Train-Of-Four score of 0 on neuromuscular monitoring, a double-lumen tube was inserted using a bronchoscope chosen by reference to the patient’s weight and height. The tube was connected to a ventilator, and general anesthesia was induced by sevoflurane. The concentration of the inhaled agent was adjusted to 0.8–1.2-fold the minimum alveolar concentration. This maintained an appropriate depth of anesthesia, as monitored by the bispectral index, which was held between 40 and 60. Remifentanil (Ultiva; GlaxoSmithKline, Brentford, UK) served as an adjunct anesthetic that was continuously administered via a syringe pump (Injectomat TIVA Agilia^®^; Fresenius Kabi AG, Bad Homburg, Germany) to maintain blood pressure within 20% of the baseline level.

After anesthesia induction, the patient was positioned in the lateral position and one-lung ventilation was initiated. All surgeries were performed by one of two surgeons, each of whom had conducted more than 100 wedge resections. Two or three thoracoscopy ports were inserted. In both groups, fentanyl 1 mcg/kg and ramosetron 0.3 mg were administered 10 min before the end of surgery to prevent nausea and vomiting. After surgery, each patient was repositioned in a supine position, the inhalation anesthetic was discontinued, and the patient was awakened before transfer to the post-anesthetic care unit (PACU), where pain was assessed.

### 2.5. Post-Surgery Analgesia

Both groups used the same PCA drug and method. The day before surgery, during the consent process, patients were informed about the benefits, potential complications, and limitations of PCA and were told to press the PCA button when their visual analog scale (VAS) pain score exceeded 4.

The PCA (Hospira Gemstar Infusion Pump; Hospira, Lake Forest, IL, USA) was fentanyl 20 mcg/kg and ramosetron 0.3 mg in normal saline. The PCA setup delivered a bolus of 3 mL (0.6 mcg/kg); the refractory period was 5 min, and the maximum dose was 40 mL (8 mcg/kg) every 4 h; continuous infusion was impossible. All PCA devices were collected at 48 h after surgery, and data were gathered.

### 2.6. Nerve Block

In the SAPB group, a nerve block was performed with the patient in the lateral position, thus before wakening from anesthesia. The block proceeded under ultrasound guidance (probe WS80A: Samsung Medison, Seoul, Republic of Korea; needle device: TaeChang Industrial, Useong-myeon, Republic of Korea). The skin was disinfected with hexitanol. After covering the probe with a sterile film, the intersection of the fifth rib and the mid-axillary line was identified. The probe was positioned perpendicular to the rib axis to show the serratus anterior, the latissimus dorsi, the ribs, and the intercostal muscles. Using the in-plane technique, the needle was advanced under ultrasound guidance, and the tip was positioned between the intercostal muscles and the serratus anterior using 0.2% ropivacaine solution. After injection of a 1 mL test dose to confirm separation of the serratus anterior muscle from the rib, the total dose was 0.4 mL/kg (Figure 1).

### 2.7. Postoperative Care

After transfer to the ward, a routine acute pain service protocol commenced. On the evening of surgery, 100 mg aceclofenac and 500 mg paracetamol were given orally. Commencing the next day, aceclofenac was given twice daily, and paracetamol was given three times daily. When the VAS pain level exceeded 4, patients were instructed to press the PCA button. If pain persisted, 50 mg tramadol was administered intravenously.

### 2.8. Study Indicators

The primary outcome was the total consumption of fentanyl during the first 8 h postoperatively. The secondary outcomes were fentanyl consumption; VAS score at rest and when coughing at 4, 8, 12, 24, and 48 h postoperatively; number of PCA requests; and complications such as hypotension caused by a central nerve block, respiratory depression, nausea, or vomiting; any additional analgesic requirement; and/or any symptom of local anesthetic toxicity such as altered consciousness and/or seizures.

### 2.9. Statistical Analysis

Statistical analysis employed SPSS software (version 22). The group means of continuous variables (VAS scores and fentanyl consumption) were compared using the t-test. The Mann–Whitney test was employed to compare ordinal categorical variables (postoperative nausea and vomiting [PONV]). The chi-square or Fisher’s exact test was used for cross-analysis of proportions.

### 2.10. Sample Size

The sample size was based on that of a study [18] that used postoperative morphine consumption to 12 h as the primary indicator. Given a standard deviation of 2.56 mg, an alpha value of 5%, a statistical power of 80%, and use of a two-tailed test, a clinically significant difference was at least 3.5 mg. Allowing for a 20% failure rate, the required sample size was 22. G*power 3.1.9.2 (Heinrich Heine University Düsseldorf, Germany) was used to calculate the sample size.

## 3. Results

Data collection began in January 2018 and ceased in March 2019. In all, 22 patients were enrolled. One patient in the SAPB group was excluded because of conversion from VATS surgery to open thoracotomy. Thus, there were ultimately 10 patients in the SAPB group and 11 in the control group (Figure 2).

There were no significant between-group differences in age, sex, height, weight, body mass index (BMI), ASA classification, anesthesia or surgery time, number of thoracoscopic ports, bleeding volume, or fluid intake (Table 1). The primary parameter, fentanyl consumption to 8 h post surgery, was significantly lower in the SAPB group than in the control group (183 ± 107 μg vs. 347 ± 202 μg, *p* = 0.035) (Table 2). In the SAPB group, fentanyl consumption was also significantly lower than that of the control group at 4, 24, and 48 h, but not at 12 h. In both groups. Although fentanyl consumption at 12 h postoperatively did not reach statistical significance, a notable difference was observed (control group vs. SAPB group: 410.5 ± 226.8 vs. 208.8 ± 141.6). The VAS scores decreased similarly over time during both resting and coughing, although the changes were not statistically significant. There were no significant between-group differences in terms of additional analgesics requested on postoperative days 1 and 2, the mean blood pressures at recovery and at 8 and 24 h post surgery, or oxygen saturation at 8 h post surgery. No patients required an antiemetic drug, and no side effects of local anesthetics were detected (Table 3).

## 4. Discussion

This randomized clinical study investigated whether a single ultrasound-guided SAPB reduced postoperative opioid consumption. The primary parameter, fentanyl consumption by 8 h post surgery, was significantly reduced in the SAPB group compared to the control group. In addition, opioid consumption fell during most of the postoperative period (to 48 h).

This manuscript highlights the precise measurement of SAPB’s efficacy by utilizing PCA devices, which provides more reliable data compared to relying solely on VAS scores.

When pain is assessed using VAS, it reflects only the pain at the specific time of measurement, making it difficult to evaluate the average pain score over 4 or 8 h. In contrast, PCA devices provide continuous data, allowing for a more accurate understanding of the average pain levels over these time periods. Furthermore, PCA measurements enable a more reliable assessment of pain during times when investigators have limited access, such as nighttime or early morning hours. Although the VAS scores were lower in the SAPB group at all time points, the differences were not significant. A simple VAS is commonly used to assess postoperative pain, but VAS scores may not measure postoperative pain reliably [19] and are linear, whereas actual pain is nonlinear [20], implying that the VAS scores may not evaluate acute pain well after surgery [21]. A VAS score reflects pain at a specific point in time, not an average pain level, and may thus not represent overall pain well. In addition, our small sample size rendered it difficult to reveal statistical significance. Opioid consumption was thus considered a better way to assess postoperative pain. Similar studies [22,23,24] have reported reduced opioid consumption after SABP.

SAPB is an effective and easily feasible procedure that does not significantly increase the overall anesthesia time. Additionally, it does not cause major complications such as respiratory depression or hypotension, making it a safe and practical option. Given these advantages, SAPB could be considered for all patients undergoing VATS to enhance postoperative pain management.

Whether a serratus plane block should be administered superficially or deeply with respect to the serratus muscle remains controversial [25,26]. One study found that superficial administration was associated with wider anesthetic spread, thus blocking the lateral branches of many intercostal nerves. Deep administration was associated with less spread [14].

If an SAPB created during VATS surgery is to be effective, the block must extend to the intercostal muscles and parietal pleura. Cadaver studies have shown that, when rib fractures are absent and local anesthetic is injected into the sub-serratus space, the continuity of the myofascial plane is not disturbed; the anesthetic does not penetrate the intercostal muscles. However, if a rib is fractured, the plane is disrupted and the anesthetic infiltrates the intercostal muscles [27]. For this reason, it has been suggested that anesthetic delivery to the deep serratus muscle may be particularly effective for cases with rib fractures [28]. In addition, many studies on patients undergoing lung surgery via thoracotomy have delivered the anesthetic to the deep serratus muscle [29,30,31]. After VATS, this may afford optimal analgesia; the local anesthetic penetrates the intercostal muscles and nerves.

CPSP is one of the most common complications after surgery, significantly impacting quality of life and imposing both economic and healthcare-related burdens. After thoracotomy, the incidence of severe CPSP is 10%, the risk of CPSP lasting for more than 12 months is 41%, and the neuropathic pain risk is 45% [32]. Appropriate nerve blocks and other multimodal approaches toward pain management reduce the CPSP risk [33,34].

During VATS surgery, a TPVB effectively controls pain by blocking one hemithorax. An erector spinae plane block is also possible [35]. However, such blocks are created with the patient in a lateral position under general anesthesia, are technically demanding, and may cause critical complications. The VATS surgical incisions are smaller than those of thoracotomy, and the port sites are clustered. SAPB can thus be performed without the need for the positional change of a patient after surgery; SABP is effective and convenient with minimal complications. EAPB and SABP may afford similar analgesic effects and reductions in opioid consumption [36].

Our study had several limitations. First, the sample size was small because the pilot study’s effect size (Cohen’s d) was 1.57, which is considered very high. Although we were able to demonstrate fentanyl consumption as the primary parameter with statistical significance, we could not achieve statistical significance for the pain relief effect in the SAPB group. With a larger sample size, it is likely that the effect of SAPB on postoperative pain could have been demonstrated more clearly. Second, the control group did not receive a sham block. SAPB involves the injection of 30 mL of local anesthetic along the fascia plane at the mid-axillary line, which may allow the anesthetic to spread to the surgical incision site. If saline had been used for the placebo block, there is a possibility that it could also spread to the incision site, potentially interfering with tissue healing and increasing the risk of infection. Although many studies attempt sham blocks, we believe the effect of a sham block on pain would be minimal and would not obscure the main findings of this study.

First, future research should compare the effectiveness of SAPB in pain management with other nerve block techniques, such as ESP block or paravertebral block, to determine its relative efficacy, including non-inferiority testing to confirm that SAPB is not less effective. Second, studies should investigate whether combining SAPB with multimodal analgesics, such as intravenous acetaminophen, enhances pain control and convenience in clinical practice. Third, given that SAPB reduces opioid demand, future studies should explore whether this also leads to a reduction in opioid-related side effects, such as nausea and vomiting.

## 5. Conclusions

This study demonstrates that SAPB significantly reduces fentanyl consumption in the postoperative period, which can minimize the risk of opioid-related side effects and complications. While statistical significance for pain relief was not achieved, the notable reduction in pain scores suggests that SAPB could improve patient recovery and comfort.

The findings of this study support the potential for SAPB to be incorporated into multimodal analgesia strategies for patients undergoing VATS, pending further validation in larger multi-center trials.

## Figures and Tables

**Figure 1 medicina-61-00011-f001:**
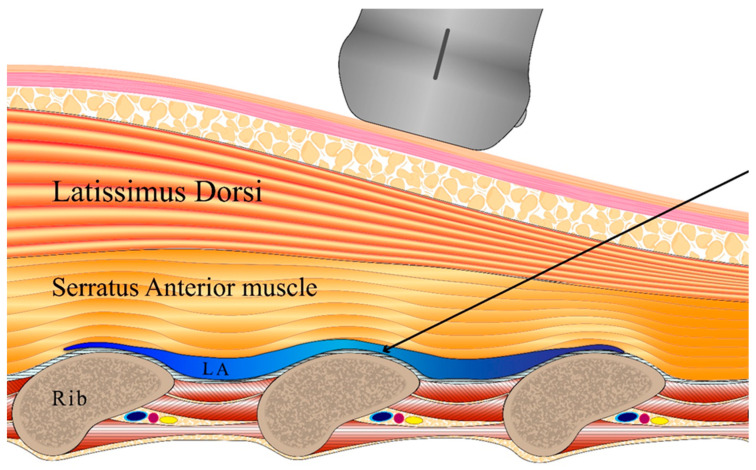
Schematic of the ultrasound-guided serratus anterior plane block using an in-plane technique. A local anesthetic diffuses between the ribs and the inferior aspects of the serratus anterior muscle. LA, local anesthetic.

**Figure 2 medicina-61-00011-f002:**
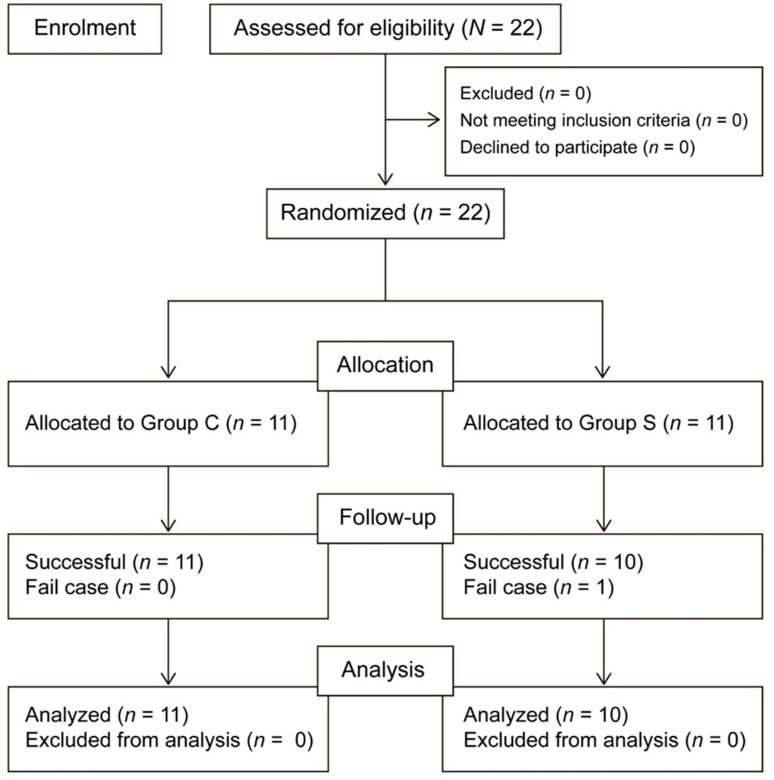
Group allocation.

**Table 1 medicina-61-00011-t001:** Patient characteristics.

	Control Group (11)	SAPB Group (10)
Age (years)	52.3 ± 19.8	46.1 ± 15.7
Sex (male, %)	8 (73%)	5 (50%)
Height (cm)	166.9 ± 8.2	166.2 ± 10.8
Weight (kg)	63.1 ± 9.1	66.5 ± 11.9
BMI (kg/m^2^)	22.6 ± 2.9	24.2 ± 4.3
Anesthesia time, min	90.5 ± 31.2	72.7 ± 20.7
Operation time, min	58.6 ± 30.6	38.2 ± 15.4
Number of ports	2.8 ± 0.4	2.8 ± 0.4
Bleeding volume, mL	40 ± 37.9	21 ± 24.2
Intraoperative fluid, mL	353.6 ± 189.4	350.5 ± 262.9
ASA status score		
1	5 (45%)	4 (40%)
2	6 (55%)	5 (50%)
3		1 (10%)
Operator		
L	6	4
S	5	6

Abbreviations: BMI, body mass index; ASA, American Society of Anesthesiologists; SAPB, serratus anterior plane block. L and S are the surgeons’ initials. Values are means ± SDs or numbers (%).

**Table 2 medicina-61-00011-t002:** Postoperative pain-related data.

	Control Group	SAPB Group	*p*-Value
Total fentanyl consumption, μg			
4 h	264.0 ± 168.9	132.6 ± 91.2	0.042
8 h	346.9 ± 202.6	183.3 ± 107.1	0.035
12 h	410.5 ± 226.8	208.8 ± 141.6	0.260
24 h	579.5 ± 294	339.9 ± 206	0.045
48 h	679.6 ± 319	383.7 ± 231.9	0.013
Visual analog pain scores Resting			
PACU	3.3 ± 0.9	2.6 ± 1.3	0.174
4 h	3.1 ± 1.4	2.5 ± 1.2	0.306
8 h	3.5 ± 1.2	2.9 ± 1	0.269
12 h	3.4 ± 1.1	2.5 ± 1.4	0.139
24 h	3 ± 1.2	2.2 ± 1	0.117
48 h	2.9 ± 1.7	1.6 ± 0.7	0.036
During coughing			
PACU	4.5 ± 0.9	3.5 ± 1.6	0.077
4 h	4.1 ± 1.5	3.6 ± 1.5	0.466
8 h	4 ± 1.4	3.1 ± 1.2	0.131
12 h	3.9 ± 1.6	2.7 ± 0.9	0.049
24 h	3.7 ± 2	2.3 ± 0.8	0.046
48 h	3.5 ± 1.8	2 ± 1.2	0.029
Tramadol administrations			
0–24 h	5	6	0.258
24–48 h	2	2	0.916

Abbreviations: SAPB, serratus anterior plane block; PACU, post-anesthetic care unit. Values are means ± SDs or numbers.

**Table 3 medicina-61-00011-t003:** Data on complications.

	Control Group	SAPB Group	*p*-Value
Number of ramosetron administrations			
0–24	0	0	N/A
24–48	0	0	N/A
Postoperative mean pressure [mmHg]			
PACU	90.1 ± 7.8	97.4 ± 17.9	0.237
8 h	81.8 ± 12	87.8 ± 12.3	0.268
24 h	84 ± 7.7	84.5 ± 12.1	0.913
SPO_2_ at 8 h post surgery	98.5 ± 0.4	98.8 ± 0.7	0.389
Continuous use of an oxygen mask	1	1	N/A
Side-effects of local anesthetics	0	0	N/A

Abbreviations: SAPB, serratus anterior plane block; PACU, post-anesthetic care unit; SPO_2_, oxygen saturation; N/A, not applicable. Values are means ± SDs or numbers.

## Data Availability

The datasets generated and/or analyzed in the current study are available from the corresponding author upon request.

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
