# Peer review of "Effects of a Serratus Anterior Plane Block After Video-Assisted Lung Wedge Resection: A Single-Center, Prospective, and Randomized Controlled Trial"

_medicina, 2024, doi:10.3390/medicina61010011_

Round 1

Reviewer 1 Report

Comments and Suggestions for Authors

At first glance, it appears to be a randomized controlled trial written on a subject on which a considerable number of studies have been conducted.

1.  The title should clearly state that the process is video-supported.

2. 'In total, 22 patients undergoing VATS lung wedge resections were randomized into two groups...' While there are studies suggesting that there should be at least 12 subjects in each group even in a pilot study, the results of a study with 11 patients in each group would be viewed with suspicion and would have little credibility.

3. What does 'Low-dose computed tomography has increased the early detection rate of lung cancer' have to do with the subject? Does a study investigating the effect of serratus block in VATS start with an easier diagnosis of lung cancer???

4. The primary outcome was stated as the fentanyl requirement at the 8th hour. When calculating the sample size, it was determined based on the 12th hour morphine consumption of the 18th reference (a retrospective study). Even though it is retrospective, these data can of course be used in determining the sample size, but a study conducted with 11 patients per group, which would be considered insufficient even for a pilot study, cannot be accepted.

5. PCA fentanyl was given while rescue analgesic tramadol was illogical. If this was done, all opioids should have been converted to morphine equivalents and statistics should have been made. Because if the given tramadol doses were taken into account, 5 doses of rescue analgesic were used in one group and 6 doses in the other. If we add this per patient, there will be 250 and 300 mg tramadol, and if these are converted to fentanyl, the statistical significance of 0.045 at 24 hours will probably turn into insignificance. It may also be the case at 4 and 8 hours.

The article is generally far from being written professionally, and the data it presents is far from being innovative.

Comments on the Quality of English Language

good

Author Response

  1. The title should clearly state that the process is video-supported.
    1. "I have revised the title as requested and added 'Video-associated.'"
  2. 'In total, 22 patients undergoing VATS lung wedge resections were randomized into two groups...' While there are studies suggesting that there should be at least 12 subjects in each group even in a pilot study, the results of a study with 11 patients in each group would be viewed with suspicion and would have little credibility.

    1. We acknowledge the concern regarding the small sample size. Initially, we aimed to include 25 subjects per group in our study. However, due to limitations set by the Institutional Review Board (IRB), the sample size was adjusted based on a previously published study [18]. That study used postoperative morphine consumption at 12 hours as the primary outcome, with a standard deviation of 2.56 mg, an alpha value of 5%, statistical power of 80%, and a clinically significant difference of at least 3.5 mg. These parameters indicated a minimum required sample size of 22 subjects, allowing for a 20% failure rate.

      Additionally, the IRB’s statistician noted that the power value in the reference study was sufficiently high, justifying the reduction in sample size. Despite our efforts, one patient in the experimental group was excluded, resulting in final group sizes of 10 and 11 patients. We recognize that this deviation might influence the credibility of our results and appreciate the reviewer's perspective.

  3. What does 'Low-dose computed tomography has increased the early detection rate of lung cancer' have to do with the subject? Does a study investigating the effect of serratus block in VATS start with an easier diagnosis of lung cancer???
    1. Sorry for any confusion caused. The statement 'Low-dose computed tomography has increased the early detection rate of lung cancer' was included to illustrate how advances in CT scan technology have made earlier diagnoses possible. This earlier detection has enabled some patients to undergo less invasive surgeries, such as wedge resection instead of lobectomy. As a result, the frequency of VATS procedures has increased. This context explains part of the background for studying the effects of serratus anterior block in VATS surgeries.

  4. The primary outcome was stated as the fentanyl requirement at the 8th hour. When calculating the sample size, it was determined based on the 12th hour morphine consumption of the 18th reference (a retrospective study). Even though it is retrospective, these data can of course be used in determining the sample size, but a study conducted with 11 patients per group, which would be considered insufficient even for a pilot study, cannot be accepted.
    1. Thank you for your comment, which I fully understand and appreciate. When we initiated this study, there was a significant lack of literature on this FICB technique. As a result, we had no choice but to rely on data from initial retrospective studies, as there were no studies investigating opioid demand over an 8-hour period. Consequently, we selected the most comparable study as a reference.

      However, as seen in the pilot study, the difference in VAS scores between the control group without nerve block at the 8-hour and 12-hour marks was nearly identical (3.6 ± 0.88 vs. 3.65 ± 0.67). Therefore, we believe that using the 8-hour measurement as an indicator would not pose significant issues.

      As I mentioned earlier, the effect size (Cohen’s d) in the pilot study was 1.57, which is considered very high. Therefore, our IRB statistician did not approve a large sample size for this study.

  5. PCA fentanyl was given while rescue analgesic tramadol was illogical. If this was done, all opioids should have been converted to morphine equivalents and statistics should have been made. Because if the given tramadol doses were taken into account, 5 doses of rescue analgesic were used in one group and 6 doses in the other. If we add this per patient, there will be 250 and 300 mg tramadol, and if these are converted to fentanyl, the statistical significance of 0.045 at 24 hours will probably turn into insignificance. It may also be the case at 4 and 8 hours.
    1. Thank you for your insightful comment. However, the reason tramadol was not included in the calculation of total opioid consumption is as follows:

      1. When tramadol was administered on the ward, it was difficult to determine the exact time of administration. Only approximate times were recorded, making it challenging to categorize the data based on specific time points.
      2. Despite thorough training, tramadol was often administered without strict adherence to the protocol of giving it only when the VAS score was 4 or higher.

      Therefore, we concluded that the most accurate measurement of opioid consumption was the value recorded directly from the PCA device post-surgery.

Reviewer 2 Report

Comments and Suggestions for Authors

Dear Colleagues! Congratulations on a great study! In my opinion, it is advisable to: 1) include the word "single-center" in the title of the article; 2) remove the word "block" in the second phrase in the "Keywords"; 3) update a number of SAPB and VATS References on the core problem, since there are only 32.4% of references for 2019-2022. There are none at all for 2023-2024. Let me ask you a few questions. 1) Your study was conducted in 2017-2019 without studying the long-term results. Why was the article not submitted earlier? 2) It would be important for readers to understand from the text of the article: 1) why was the exclusion criterion "anticoagulant therapy or blood clotting disorders", while the study included not only SAPB pain relief, but also VATS surgeries? (line 90); 2) what were the signs of "apparent muscle relaxation" (line 113)? 3) why was there no difference in fentanyl consumption in the two groups 12 hours after surgery? (line 196); 4) why, although not statistically significant, did the duration of surgery differ in the two groups (Table 1)?

Author Response

  1. include the word "single-center" in the title of the article; 
    1. "Yes, I have made the changes as you suggested."

  2. remove the word "block" in the second phrase in the "Keywords";

    1. Yes, I have made the changes as you suggested."
  3. update a number of SAPB and VATS References on the core problem, since there are only 32.4% of references for 2019-2022. There are none at all for 2023-2024. Let me ask you a few questions. 1) Your study was conducted in 2017-2019 without studying the long-term results. Why was the article not submitted earlier?
    1. Thank you for your insightful comment. I agree that publishing the results earlier would have gained more recognition. However, at that time, the hospital workload was overwhelming, and I had numerous responsibilities, leaving me little time to focus on the research. It was only after recently transitioning to a new hospital that I was able to dedicate time to complete the manuscript.

  4. It would be important for readers to understand from the text of the article: 1) why was the exclusion criterion "anticoagulant therapy or blood clotting disorders", while the study included not only SAPB pain relief, but also VATS surgeries? (line 90);
    1. All nerve blocks carry a risk of hematoma due to vascular puncture. While most hematomas are reabsorbed without significant issues, some may lead to nerve damage due to compression. For this reason, nerve blocks are often skipped in cases of coagulation disorders. Although the serratus anterior plane block has a very low risk of such complications, at the time of the study, limited research was available. Therefore, anticoagulant therapy and clotting disorders were set as exclusion criteria for safety.

      We will add the following explanation to the manuscript: ‘To prevent hematomas caused by vascular puncture.

  5. what were the signs of "apparent muscle relaxation" (line 113)? 
    1. I will add the following statement.

    2. Once the loss of consciousness was adequate and muscle relaxation was apparent, as confirmed by a TOF score of 0 on neuromuscular monitoring, 
  6. why was there no difference in fentanyl consumption in the two groups 12 hours after surgery? (line 196); 
    1. Thank you for your insightful comment. Although the difference was not statistically significant, there was a substantial difference in fentanyl consumption between the two groups (410.5 mg vs. 208.8 mg). We ran the analysis multiple times using SPSS to confirm the results, and they remained consistent. We believe this lack of statistical significance is likely due to the small sample size. Despite the lack of statistical significance, we do not think this negates the clinical effect of the nerve block.

      We will add the following statement to the manuscript:
      "Although fentanyl consumption at 12 hours postoperatively did not reach statistical significance, a notable difference was observed (control group vs. SAPB group: 410.5 ± 226.8 vs. 208.8 ± 141.6)."

  7. why, although not statistically significant, did the duration of surgery differ in the two groups (Table 1)?
    1. I agree that this observation is unusual. Generally, longer surgeries tend to involve more extensive procedures, which can result in greater postoperative pain. If the surgery duration had been shorter in the control group, it might raise concerns about intentional researcher bias. However, since the surgery duration was longer in the control group, we believe this does not undermine the main findings of the study.

Reviewer 3 Report

Comments and Suggestions for Authors

Dear authors, thanks for the opportunity to revise your paper entitled "Effects of a serratus anterior plane block after lung wedge resection: A prospective, randomized controlled trial". Authors have performed RCT on the use of serratura plane block after lung wedge resection. Even if the study seems well conducted, given also the small sample size, it does not add anything to already existing literature, due to the fact that SAPB is a well established fascial block in thoracic surgery. Following my concerns:

- English should be revised with the help of a native speaker to improve clarity and fluency.

- Sample size: even if authors describe the methods they use for sample size, it is very small and based only on one study. Moreover, they used a 3.5mg reduction in opioid consumption as clinical significant effect size, but they did not justified it. Please explain. The article does not mention any software used for the calculation, which would have been helpful to confirm the accuracy of the analysis.

- Were factors such as comorbidities, preoperative pain levels, or other confounding variables accounted for? These could influence postoperative pain and might need to be included in the analysis.

- The control group did not receive a placebo block, and this can be a potential bias. 

- Why BMI as exclusion criteria?

Comments on the Quality of English Language

English should be improved through the entire manuscript.

Author Response

Following my concerns:

  1. English should be revised with the help of a native speaker to improve clarity and fluency.
    1. I am not a native speaker, so it is understandable that my English manuscript may sound awkward at times. For this reason, I used a professional proofreading service, and I can provide you with the certificate of proofreading upon request. If you could kindly point out specific areas that appear awkward, I will have them revised again. I apologize for any inconvenience caused and appreciate your understanding.

  2.  Sample size: even if authors describe the methods they use for sample size, it is very small and based only on one study. Moreover, they used a 3.5mg reduction in opioid consumption as clinical significant effect size, but they did not justified it. Please explain.

    1. According to a recent study titled "Minimal clinically important differences in randomized clinical trials on pain management after total hip and knee arthroplasty: a systematic review," the minimal clinically important difference (MCID) for 0-24 hours was found to be 10 mg. Therefore, using 3.5 mg as the clinically significant effect size for an 8-hour period seems to be an appropriate and reasonable value.

  3. The article does not mention any software used for the calculation, which would have been helpful to confirm the accuracy of the analysis.

    1. Thank you for your valuable feedback. I will add the following statement to clarify:

      "Gpower (Heinrich Heine University Düsseldorf, Germany) was used to calculate the sample size."

  4. Were factors such as comorbidities, preoperative pain levels, or other confounding variables accounted for? These could influence postoperative pain and might need to be included in the analysis.
    1. Patients with chronic preoperative pain or drug dependency were excluded from the study. All participants were in a state without significant pain preoperatively. Patients with a bleeding tendency, those who refused nerve block, or those with psychiatric disorders preventing accurate communication were also excluded. Additionally, participants were excluded if the extent of surgery exceeded what is typically considered VATS, such as cases with bleeding exceeding 1L or surgery duration over 6 hours.

  5. The control group did not receive a placebo block, and this can be a potential bias. 
    1. This is an excellent point. As noted in the limitations section, a sham block was not performed for the control group. In future studies, we will ensure to include a sham block.

      SAPB involves the injection of 30 ml of local anesthetic along the fascia plane at the mid-axillary line, which may allow the anesthetic to spread to the surgical incision site. If saline had been used for the placebo block, there is a possibility that it could also spread to the incision site, potentially interfering with tissue healing and increasing the risk of infection.

    2. I will include this point in the limitations section.
  6. Why BMI as exclusion criteria?
    1. BMI was included as an exclusion criterion because, during the study planning, it was considered that higher BMI might place the target structures for SAPB at deeper locations, making them less visible on ultrasound. This could potentially result in ineffective nerve blockade.

Round 2

Reviewer 1 Report

Comments and Suggestions for Authors

Unfortunately, the methodology, sample size and data presented in this study are unacceptable. This study can only be a case series. Even the authors' responses to the referees reveal how carelessly they wrote. For example, 'When we initiated this study, there was a significant lack of literature on this FICB technique.' What kind of a response is this? Also, the sample size and power analyses are far from scientific. I cannot accept this manuscript.

Author Response

Dear Reviewer,

Thank you for taking the time to provide detailed feedback on our manuscript. I sincerely value your input and the opportunity to address the concerns you have raised.

  1. Methodology and Sample Size:
    I understand your concern regarding the sample size and power analysis. When we initiated this study, we faced significant challenges due to the limited literature available on the FICB technique, which made it difficult to establish strong reference points for sample size calculation. Nonetheless, we used the best available data and adhered to standard methods to determine a feasible sample size. We recognize the limitations of a smaller sample and have acknowledged this in the manuscript, including a discussion of its impact on statistical power and generalizability.

  2. Responses:
    I regret that one of my previous responses was perceived as careless. My intention was to provide context regarding the study's initiation rather than dismiss the concern. I understand the importance of precise and thoughtful communication, and I will ensure that all responses and manuscript revisions are further refined for clarity and professionalism.

Once again, I am grateful for your thorough review, which has helped us identify areas for improvement. I am committed to addressing these issues comprehensively to enhance the quality of our manuscript.

Reviewer 3 Report

Comments and Suggestions for Authors

Thanks for the revision.

The quality of the manuscript has now improved, even if it lacks of novelty and add very little to already existing literature. 

Author Response

Dear Reviewer,

Thank you for your thoughtful review and for acknowledging the improvements made to the manuscript. I understand and agree with your perspective that the novelty of this study may be limited and that it adds incrementally to the existing literature.

However, I believe this work serves as an important stepping stone for future research. By addressing key aspects of SAPB in VATS patients, it provides a foundation upon which more comprehensive studies can be built. This manuscript lays the groundwork for exploring additional areas, such as comparisons with other nerve block techniques or the incorporation of multimodal analgesia strategies, which we have proposed as future research directions.

I deeply appreciate your constructive feedback and the opportunity to refine this manuscript further.

Best rewards,

Woojin Kwon